# Alternate Ultrasound/Microwave Digestion for Deep Eutectic Hydro-distillation Extraction of Essential Oil and Polysaccharide from *Schisandra chinensis* (Turcz.) Baill

**DOI:** 10.3390/molecules24071288

**Published:** 2019-04-02

**Authors:** Jun-Han Li, Wei Li, Sha Luo, Chun-Hui Ma, Shou-Xin Liu

**Affiliations:** College of Material Science and Engineering, Northeast Forestry University, Harbin 150040, China; nefulijunhan@163.com (J.-H.L.); liwei19820927@126.com (W.L.); luo.sha.85@163.com (S.L.)

**Keywords:** deep eutectic solvent, ultrasound/microwave digestion, *Schisandra chinensis* (Turcz.) Baill, essential oil, polysaccharide

## Abstract

An alternating synergetic ultrasound/microwave method was applied to the simultaneous extraction of essential oils and polysaccharides with deep eutectic solvent (DES) from *Schisandra chinensis*. Under the optimal conditions, extract in the selected choline chloride-ethylene glycol 1:3 solvent yielded 12.2 mL/kg and 8.56 g/100g of essential oils and polysaccharides, respectively. The free radical scavenging and immunological activities of the polysaccharides and the antioxidant activity of the essential oils have also been investigated. The lymphocyte proliferation capacity was substantially improved by adding concanavalin A or lipopolysaccharides to polysaccharides (0.20 mg/mL). The IC50 values of the essential oils for scavenging DPPH obtained by hydro-distillation and DES ultrasound/microwave-assisted hydro-distillation (DES UMHD) were 52.34 µg/mL and 29.82 µg/mL, respectively. The essential oil obtained by DES UMHD had the highest reducing power (856.05 (TE)/g) at 150 g/mL and had the strongest inhibitory capacity (SC% = 18.12%). *S. chinensis* has the potential to be developed as a natural antioxidant.

## 1. Introduction

Deep eutectic solvents (DESs) are a sustainable alternative to ionic liquids derived from green natural and renewable components. Compared with the traditional extraction methods, DES pretreatment can enhance digestibility, reduce energy consumption, and simplify procedures for solvent recovery. DES can be formed by mixing simple quaternary ammonium salt such as choline chloride (ChCl) with appropriate Hydrogen Bond Donor (HBD) (such as metal halides, acids and alcohols) under heating [1]. These prepared DES are non-toxic, easily synthetic, and have low cost of preparation. There are reports on the applications of DESs for the extraction of phenolic compounds [2], polysaccharides [3], and proteins [4] from plants, and research continues to explore more applications.

The study of different assisted methods, such as ultrasound-assisted, which use the mechanical, cavitation, and thermal effects of ultrasound to increase penetration of the medium by increasing the velocity of its motion [5], has been conducted in recent years. The cavitation produced by ultrasound greatly improves the heterogeneous reaction rate, achieves uniform mixing between heterogeneous reactants, accelerates reactant and product diffusion, promotes the formation of new solid phases, and controls the size and distribution of particles [6,7]. In contrast, microwave irradiation is as an energy source to induce molecular movement and rotation of liquids with permanent dipoles [8]. Thus, the biomass material absorbs microwave energy and cell temperature increases rapidly, such that the internal pressure of the cell exceeds the expansion capacity of the cell wall, resulting in cellular breakdown. This allows intracellular active ingredients to flow freely and be captured and dissolved at lower temperatures. Thus, it has a wide application range [9,10].

Alternate ultrasound/microwave extraction technology, which directly combines the two extraction modes to make full use of both ultrasound vibrational energy and thermal effect of microwave irradiation [11]. This method has many advantages, such as fast heating speed, low energy consumption, low solvent requirements, and high recovery rate. Simultaneous ultrasound/microwave extraction is also beneficial for the extraction of polar and thermally unstable components and avoids the decomposition that results from extraction or synthetic procedures with long processing times, high temperatures, and high pressures [12]. This technique has been applied to the extraction of alkaloids [13], flavonoids [14], polysaccharides [15], and other active substances from plants.

*Schisandra chinensis* (Turcz.) Baill (*S. chinensis*) polysaccharides have attracted the attention of researchers with its liver protecting, kidney reinforcing, anti-fatigue, anti-oxidative, immunity enhancing, antiaging, tumor inhibition, and anti-diabetic effects [16,17]. *S. chinensis* essential oils are extracted from plant-specific aromatic substances and can prevent cough, promote DNA synthesis, and be an inhibitor of plasminogen activation [17,18]. The production of natural ingredients requires a preliminary study of compound separation and for optimal extraction methods and conditions to be established [19]. Although a variety of solid-liquid extraction methods can be used to obtain natural ingredients, these procedures have shortcomings, including long extraction times, large solvent requirements, and partial loss of natural molecules, such as phenolic compounds.

In the field of food chemistry, simultaneous ultrasound/microwave is often used to extract natural products. In contrast, alternating ultrasound/microwave extraction technology has rarely been reported [15]. However, the efficiency of synergetic ultrasound/microwave extraction is not necessarily better than that of alternating ultrasound/microwave digestion, and the extraction efficiency of natural products is greatly influenced by combining extraction modes in an alternating fashion [14]. Therefore, this work aimed to develop an alternating ultrasound/microwave extraction method with DES that simultaneously extracts higher yields of polysaccharides and essential oils to study the optimal combination of alternating ultrasound/microwave extraction methods, and analyzed its mechanism. The extraction mechanisms of the alternating ultrasound/microwave methods are also explained in detail. After deproteinization, free radical scavenging activity and cell proliferation of the polysaccharides, the antioxidant activities of the essential oils were investigated [20,21]. This also demonstrates that *S. chinensis* polysaccharides and essential oils might be used as potential natural antioxidants and immunostimulants that conform to the principles of sustainability and green chemistry.

## 2. Results and Discussion

### 2.1. Screening of DES

In a typical reaction, ChCl (hydrogen bond acceptor) and ethylene glycol (EG), glycerol (GI), or 1,4-butanediol (BDO) (hydrogen bond donors) were mixed. A greater number of OH is provided by GI when the ratios of polyol are the same; however, better water absorbent performance of GI may form intermolecular hydrogen bonds, resulting in a decrease in the number of hydrogen bonds with ChCl. Compared with BDO, steric effected the formation of intermolecular hydrogen bonds with smaller ChCl. Therefore, the screening optimization of DES HBA and HBD is of great significance. In our study, ChCl and EG, GI or BDO were mixed separately in different molar ratios (1:1, 1:2, 1:3, 1:4, 1:5), and heated at 80 °C oil bath until a transparent solution was obtained. Then, water was added to maintain a solution ratio (DES:H_2_O) of 7:3 *v*/*v*. The polysaccharide extracted by using various DESs is presented in Figure 1a. The results show that the DES solvent (ChCl-EG 1:3 molar ratio) extracted chitin showed a high yield of 6.86 ± 0.10 g/100g. After that, the DES was selected as ChCl-EG 1:3, and the solution ratios (5:5, 6:4, 7:3, 8:2, 9:1) were changed to obtain an optimal condition of 7:3 *v*/*v*. In addition, the viscosity of the solution with different solution ratios was measured—it increased with the increase of solution ratio, especially after 7:3 *v*/*v*, and the two showed a good nonlinear correlation. The viscosity standard curve followed the formula *Y* = −102.88 + 486.13*X* − 744.75*X*^2^ + 389.24*X*^3^ (*R*^2^ = 0.9996) (Figure 1b). Therefore, ChCl-EG 1:3 (7:3 *v*/*v*) was selected as DES for subsequent experiments.

### 2.2. Single Factor Analysis of Polysaccharide Extraction

#### 2.2.1. Kinetic Study of Polysaccharide Extraction

Using reflux extraction, the extraction yield of *S. chinensis* polysaccharides tended to be balanced after 4 h, reaching 8.56 g/100g (dry weight). In the ultrasound extraction process, the polysaccharide yield increased with increasing extraction time (Figure 2), but showed little further variation after an extraction time of 40 min, remaining at 7.44 ± 0.18 g/100g. In the microwave extraction process, the polysaccharide yield increased with increasing extraction time, but showed almost no further change after an extraction time of 20 min, with the polysaccharide yield remaining at 8.02 ± 0.26 g/100g. Therefore, the optimal extraction times for the extraction methods were 4 h, 40 min, and 20 min for reflux, ultrasound-assisted, and microwave-assisted extraction, respectively.

#### 2.2.2. Solid–Liquid Ratio for Extracting Polysaccharides

Using the reflux extraction method to optimize the solid-liquid ratio for polysaccharide extraction, five *S. chinensis* dry powder samples (each 10.0 g) were added to distilled water at solid-liquid ratios of 1:10, 1:20, 1:30, 1:40, and 1:50 *w*/*v*, refluxed for 4 h, and the polysaccharide extraction rates calculated (Figure 3a). When the solid-liquid ratio was increased from 1:10 to 1:30, the polysaccharide extraction rate increased. Solid-liquid ratios of more than 1:30 resulted in lower polysaccharide extraction rates. Therefore, the optimal solid-liquid ratio was 1:30 (*w*/*v*).

#### 2.2.3. Power of Ultrasound and Microwave Treatments for Polysaccharide Extraction

Using the optimal conditions of *S. chinensis* dry powder (10.0 g) in distilled water at a 1:30 (*w*/*v*) ratio, the extractions were done using different amounts of ultrasound and microwave power. Single factor experiments were performed for ultrasound and microwave power (Figure 3b,c). At ultrasound powers of less than 550 W, the polysaccharide yield increased with increasing ultrasound power. However, the polysaccharide yields at ultrasound powers of 550 W and 700 W were similar. Therefore, an ultrasound power of 550 W was selected for subsequent experiments. In the microwave extraction process, the polysaccharide yields gradually increased with increasing microwave power, with 250 W selected as optimal microwave power for subsequent experiments.

### 2.3. Optimization Extraction Parameters of Polysaccharides by Response Surface Method (RSM)

#### 2.3.1. Optimization Parameters for Ultrasound-assisted Extraction (UAE)

To further study the interactions between the factors, we optimized the solid–liquid ratio, extraction time, and ultrasound power. In Table 1, the maximum value of extraction efficiency was defined as 100%, and compared with the other values. The model F-value of 19.51 indicated that the model was significant, but the probability for the occurrence of such model F-value was only 0.01%, which can be treated as statistical noise. Values of “Probability > F” less than the 0.0500 indicated model terms were significant. The “lack of fit F-value” of 247.97 implied that the lack of fit was significant. The probability of the occurrence of such a “lack of fit F-value” was only 0.01% and can be treated as statistical noise. The response surfaces for the effect of independent variables on extraction efficiency of lignans are shown in Figure 4. Results of the test of significance for regression coefficient for ultrasound-assisted extraction is given in Appendix A.

The final extraction efficiency (*Y*) was given by *Y* = 7.75 + 1.67*A* + 0.49*B* + 0.37*C*. The software predicted that the yield of polysaccharides was 8.44 g/100g; however, in our realistic operation process, the parameters of ultrasound power, extraction time and solid-liquid ratio were selected as 550 W, 40 min, and 1:30 respectively; the yield of polysaccharides was 8.56 g/100g, and error rate was 1.42%.

#### 2.3.2. Optimization Parameters for Microwave-assisted Extraction (MAE)

To further study the interactions between the factors, we optimized the solid–liquid ratio, extraction time, and microwave power. In Table 2, the maximum value of extraction efficiency was defined as 100%, and compared with the other values. The model F-value of 17.30 indicated that the model was significant, but the probability for the occurring of such model F-value was only 0.01%, which can be treated as statistical noise. Values of “Probability > F” less than the 0.0500 indicated model terms were significant. The “lack of fit F-value” of 47.86 implied that the lack of fit was significant. The probability of the occurrence of such a “lack of fit F-value” was only 0.03% and can be treated as statistical noise. The response surfaces for the effect of independent variables on extraction efficiency of lignans are shown in Figure 5. Besides, all results of the test of significance for regression coefficient for microwave-assisted extraction can be checked in the Appendix A.

The final extraction efficiency (*Y*) was given by *Y* = 8.33 + 1.33*A* + 0.50*B* + 0.41*C* + 0.23*AB* + 0.18*AC* + 0.054*BC* − 0.35*A*^2^ − 0.45*B*^2^ − 0.15*C*^2^. The software predicted the yield of polysaccharides was 8.33 g/100g, however, in our realistic operation process, the parameters of microwave power, extraction time and solid-liquid ratio were selected 250 W, 20 min and 1:30 respectively, and the yield of polysaccharides was 8.44 g/100 g, the error rate was 1.32%.

### 2.4. Alternate Ultrasound/Microwave Extraction Design

Based on the above optimum experiments, the experimental arrangement of alternating ultrasound and microwave-assisted extractions of polysaccharides was designed, and the essential oil yield was also detected. The results showed that microwave-assisted extraction obtained a higher polysaccharide yield (7.83 ± 0.12 g/100g compared with 7.02 ± 0.11 g/100g for ultrasound-assisted extraction). In alternating experiments C–F, with extraction time under the same conditions, the polysaccharide yield obtained by microwave extraction was slightly higher than that by ultrasound treatment at first, with a higher number of alternations leading to a shortened extraction time and higher polysaccharide yield (Figure 6).

Important factors for evaluating extraction methods are extraction rate, cost, ease of operation, time, and environmental pollution. Herein, the alternating ultrasound and microwave extraction methods were evaluated with respect to the yield, extraction time, and environmental assessment of the crude polysaccharides. Refluxing extraction requires an extraction time of 4 h and uses a heating sleeve that consumes 1 kW of power every hour, giving a total energy consumption of 4.0 kWh, and emits around 3200 g of carbon dioxide. However, on using microwave heating power of 0.25 kW and ultrasound auxiliary power of 0.55 kW, the alternating extraction method emitted only 360 g of carbon dioxide per experiment. Alternating experiment F achieved the highest polysaccharide extraction rate of 8.87 ± 0.29 g/100g (Table 3).

In the electric field generated by microwaves, friction occurs between the moving ions and the medium, which may cause heating; besides, dipolar molecules attempt to align with alternating electric fields in the medium, and their oscillations cause collisions with surrounding molecules to generate heat. As the temperature inside the cell rises rapidly, the intracellular pressure increased beyond the capacity tolerated by the expansion of the cell wall, resulting in cell rupture, which accelerated the rate at which the polysaccharide molecules of *S. chinensis* diffused from the solid to the solid-liquid interface. Then, at sufficiently high power, the sparse cycle produced by the ultrasound might exceed the attractive force of the molecules of the fluid and form cavitation bubbles. The suspended solid promotes the collapse of the asymmetric bubble, which produces extremely high-speed jets that are directed at *S. chinensis*, thereby promoting its extraction.

Using SEM to observe the fruit before and after different extraction methods, it was concluded that ultrasound expanded and ruptured the cells through shaking and cavitation, while microwave destroyed most of the cell walls in the alternating treatment, which greatly improved the rate of heterogeneous reaction and achieving uniform mixing between the heterogeneous reactants (Figure 7). As such rupture has rarely been observed using ultrasound, it is conceivable that essential oils permeated the cuticle to be extracted during ultrasound treatment.

The schematic representation of the analysis mechanism of optimal alternating ultrasound and microwave-assisted extraction of *S. chinensis* polysaccharides is shown in Figure 8. The dried fruit of *S. chinensis* has a high degree of lignification and the surface of the raw material is dense. The first step is microwave-assisted extraction—the microwave selectively heats the intracellular polar molecules, such that the intracellular pressure of the cells is larger than the extracellular pressure; and ruptures and some effective substances are dissolved from the cells. In the ultrasound method, due to its cavitation effect, the bubble mix in the extracellular solvent collapses, and the jet is formed outside the cell to increase the flow rate of the solvent on the cell surface and enter the cell interior. In the microwave treatment (third step), the thermal effect not only cause the cells to swell again, but also rapidly transfers the solvent of the dissolved polysaccharide to the extracellular, which greatly increases extraction efficiency and allows more effective substances to be dispersed outside the cells. Therefore, when the fourth step uses more ultrasound, the permeability of the cell membrane is further increased due to the oscillating effect, and the instantaneous large pulling force generated by the gas blasting causes more solvent to enter the cell; the solvent and gas dissolved in the solvent also enter the cell, and bubble collapse occurs, such that the effective substance is better dissolved in the solvent.

In summary, alternate ultrasound/microwave extraction treatment destroys the sample microstructure without a high-temperature heat source, which could eliminate the temperature gradient and heating speed, while avoiding decomposition and isomerization of natural products due to long extraction times. Therefore, compared with reflux extraction, this method is more conducive for the extraction of heat-sensitive substances. It is a more efficient extraction method in the pretreatment of plant material in the food chemistry field.

### 2.5. Polysaccharide Deproteinization

An increasing amount of deproteinized proteins causes the protein content in the crude polysaccharide solution to gradually decrease, and the polysaccharide content to diminish (Figure 9). In the first four deproteinizations, the protein content reduction ratio and polysaccharide loss were larger. After four rounds, the protein and polysaccharide content slowly decreased. In order to ensure polysaccharide content, deproteinization was performed no more than four times. The deproteinized polysaccharide solution was concentrated and lyophilized to obtain refined *S. chinensis* polysaccharides in a yield of 5.11 g/100 g *S. chinensis*. This process eliminates DES residues and completely separates DES from the polysaccharides.

### 2.6. Free Radical Scavenging Activity of Polysaccharides

Excellent antioxidant activity of *S. chinensis* polysaccharide was demonstrated in hydroxyl radical scavenging tests (Figure 10). The hydroxyl radicals were produced by Fenton reaction. Hydrogen peroxide/Fe^2+^ system can produce free radicals as follows: H_2_O_2_ + Fe^2+^ → •OH + OH^–^ + Fe^3+^. It can be seen that the generation of a hydroxyl radical depends on the content of Fe^2+^ and H_2_O_2_. The polysaccharide scavenges the excessive oxygen free radicals produced in vivo and blocks the free radical reaction chain in vivo. The polysaccharide was formulated to a concentration of 0.5 mg/mL, and serially diluted to 0.4, 0.3, 0.2, 0.1 mg/mL. Polysaccharide sample solutions and vitamin C (VC) solutions with the same concentrations were prepared. At low concentrations (less than 0.20 mg/mL), the polysaccharide free radical scavenging activity was similar to that of VC. At a concentration of 0.20 mg/mL, the free radical scavenging activities were 28.02% (polysaccharide) and 23.68% (VC). However, at sample concentrations of more than 0.20 mg/mL, the scavenging activity of the polysaccharide increased with increasing concentration, while the hydroxyl radical scavenging activity of VC gradually plateaued. At a concentration of 0.50 mg/mL, the clearance rates using the polysaccharides and VC were 66.3% and 36.49%, respectively. Therefore, the hydroxyl radical scavenging activity of *S. chinensis* polysaccharides was better than that of VC at sample concentrations of more than 0.20 mg/mL. This was because according to the Fenton reaction, there are three reagents—oxidant, reducer and metal ion—and the generation of a hydroxyl radical depends on the content of Fe^2+^ and H_2_O_2_. However, there are several nature metal ions (such as Fe, Zn, Cu) in polysaccharide. Moreover, if the absolute concentrations of Fe^2+^ are higher than H_2_O_2_ and VC, a part of VC was used to reduce Fe^3+^, which results in inhibition of H_2_O_2_. Finally, the Fenton reaction was inhibited and •OH was not generated [22,23]. Therefore, the hydroxyl radical scavenging activity was better at higher concentrations of polysaccharides compared to VC.

### 2.7. In vitro MTT Activity Test of Polysaccharides

*S. chinensis* polysaccharides have strong immune activities that can significantly promote the proliferation of mouse splenic lymphocytes, with increasing polysaccharide concentration also increasing the cell proliferation rate (Table 4). At a polysaccharide concentration of 0.20 mg/mL, the cell proliferation rate was the largest, at about 30%. When the polysaccharide concentration reached 0.80 mg/mL, the cells showed almost no proliferation, with a proliferation rate of less than 5.00%. Therefore, 0.20 mg/mL was selected as the optimal *S. chinensis* polysaccharide concentration to promote cell growth. 

ConA and Lps are effective mitogens that promote lymphocyte transformation. This experiment confirmed the synergistic effect of *S. chinensis* polysaccharides and ConA or Lps cells. The role of ConA and Lps was to proliferate more than twice as much as the polysaccharides (Table 5). Meanwhile, adding polysaccharides and ConA or Lps cells significantly increased lymphocyte proliferation capacity. The synergistic effect of *S. chinensis* polysaccharides with the Lps cells was about 1.5 times better than that with the ConA cells.

### 2.8. Kinetic Study of Essential Oils

The traditional hydro-distillation method is used to extract essential oil; essential oils precipitates with boiling water at 100 °C, and the polysaccharide is extracted by aqueous extraction-alcohol precipitation method. Thus, the DES hydro-distillation method was suitable for simultaneously extracting crude polysaccharides and essential oils, and the crude polysaccharide was precipitated by subsequently increasing the alcohol content of the DES solution, and then filtered, which completely separated polysaccharide from DES solution. When the extraction time of MHD and DES UMHD treatment was increased from 0 min to 20 min, the average essential oil extraction efficiency increased dramatically. Therefore, 20 min was selected as the extraction time for further experiments. The essential oil yield showed little change after 25 min of MHD and DES UMHD treatment. However, in the HD reaction, essential oil precipitation slowly began to appear after 30 min, with 120 min required to reach the essential oil yield obtained with the other methods, and further treatment resulted in a small steady increase (Figure 11). This phenomenon, which is attributed to heat transfer, is mainly achieved through conduction and convection only in the HD process. In contrast, in the MHD and UMHD processes, heat transfer was achieved through radiation, conduction, and convection. The ionization of DES in water produced salts that make the essential oils more likely to precipitate. Among them, DES UMHD produced the highest essential oil yield. As for ChCl-based acidic DESs, studies have inferred that Cl^-^ of ChCl (HBA) with strong H^-^ bond acceptability could help to disrupt the intermolecular hydrogen bonding network of biomass and facilitate its dissolution, which is benefit for the subsequent HBD access and attack on the acid catalytic sites of biomass. Thus, better biomass deconstruction and hydro-distillation performance of DESs solution was achieved as compared to the water, thereby increasing the yield of the essential oil [24].

### 2.9. Antioxidant Analysis of Essential Oils

In our study, the antioxidant capacity of *S. chinensis* essential oils obtained using different extraction methods was studied in vitro using three complementary test systems, namely DPPH, FRAP, and β-carotene scavenging activity assays.

Freshly prepared DPPH solution has a deep purple color with maximum absorption (1.24) at 517 nm that generally fades when an antioxidant is present in the medium. Therefore, antioxidant molecules can annihilate hydrogen atoms or electrons provided by DPPH and convert them into colorless products, resulting in a decrease in absorbance at 517 nm. With increasing essential oil concentration, a significant increase in inhibition rate was observed, with the low IC50 value indicating that the sample had strong antioxidant activity. The IC50 values of DES UMHD and HD samples were 29.82 µg/mL and 52.34 µg/mL, respectively (Figure 12a). In summary, the DPPH scavenging effect was reduced in the order VC > BHA > DES UMHD > HD. In this study, the antioxidant activity of essential oils has shown that monoterpenes have a higher antioxidant effect, while the major component of *S. chinensis* essential oil were sesquiterpenes. Therefore, in my opinion, the lack of monoterpenes in essential oil was one of the possible reasons for its weak antioxidant activity [25]. This result indicated that DES UMHD extracts more essential oils, which had a more pronounced effect on DPPH than HD.

The reduction capacity of a compound can indicate its potential antioxidant activity. A higher absorbance indicates a higher ferric-reducing power, with the essential oils from different extraction methods and standards showing increasing ferric-reducing power as the concentration increased (Figure 12b). At 0.15 mg/mL, the highest reducing power of DES UMHD (A_593_ = 1.817) was 856.05 (TE)/g, while the reducing power of HD (A_593_ = 0.365) was 130.05 (TE)/g. Therefore, the reducing power of essential oils and antioxidants was in the order of DES UMHD > VC > BHA > HD.

Among the essential oils prepared by different methods, the essential oil resulting from UMHD had the highest inhibition capacity (SC% = 18.12%) in Figure 12c. The inhibition capacity of linoleic acid oxidation by HD was 14.25%. Therefore, for the β-carotene-linoleic acid bleaching assay, a descending order of inhibition capacity of VC > DES UMHD > HD > BHA is shown.

### 2.10. Chemical Composition of S. chinensis Essential Oil by Different Extraction Methods

The compositions of *S. chinensis* essential oil were obtained through HD, MHD and DES UMHD. Thirty-six components of essential oil obtained by DES UMHD and 18 components obtained by HD and MHD were investigated. The relative amount percent of individual volatile compounds was expressed as percent peak areas relative to total peak areas by GC-MS analysis (RA %) (Table 6). The main components were the same—Ylangene, (*E*)-α-bergamotene and β-himachalene—above 50% of the total peak area (Figure 13). Olefin had high unsaturation and strong reducing ability. As can be seen from Figure 13, the olefin content was relatively high—Ylangene (RA = 28.63%), (E)-α-bergamotene (RA = 10.62%), and β-himachalene (RA = 10.60%)—thus, showing that the essential oil had a certain antioxidant capacity.

## 3. Materials and Methods

### 3.1. Raw Materials and Chemical Reagents

*S. chinensis* fruit was purchased in October 2017 from Three Trees Market in Harbin City, Heilongjiang Province, China. Mature fruit was picked in September 2017 in the Greater-Lesser Khingan Mountains region of Northeast China and identified by Professor Yongzhi Cui of Northeast Forestry University (Harbin, China) as *S. chinensis* fruit originating from Heilongjiang Province. The *S. chinensis* fruits were air-dried and sieved. The resultant 20–40 mesh powder was cryopreserved at −20 °C, and uniformly mixed before sampling. Choline chloride (ChCl) was purchased from Macklin Biochemical Co., Ltd. (Shanghai, China). The chemical reagents used in this experiment, Coomassie brilliant blue G250, bovine serum albumin (BSA), fetal calf serum, dimethyl sulfoxide (DMSO), 2,2-Diphenyl-1-picrylhydrazyl (DPPH) and vitamin C (VC), were purchased from Sigma-Aldrich (Louis, Missouri, USA), and the other reagents were purchased from Tianjin Kemiou Chemical Reagent Co., Ltd. (Tianjin, China). All solvents and chemicals were analytical grade.

### 3.2. Preparation of DES

Because ChCl has strong water absorption, it was placed in a vacuum oven at 80 °C for 24 h before the experiment. A certain amount of ChCl and EG was weighed and placed into a conical flask; the composition of DES was uniformly mixed according to the molar ratio, and the specific molar ratios (1:1, 1:2, 1:3, 1:4, 1:5) were operated [4], heated, melted in an 80 °C oil bath, and magnetically stirred until it became a colorless clear liquid—ChCl-EG mixed with a given volume of water before use. Similarly, ChCl and GI, ChCl and BDO were weighed to obtain ChCl-GI and ChCl-BDO.

### 3.3. Methods

#### 3.3.1. Polysaccharide Extraction Methods

##### Determination Method of Polysaccharide

The phenol-sulfuric acid method uses sulfuric acid to hydrolyze polysaccharides into monosaccharides, followed by rapid dehydration to result in aldehyde derivatives that form orange-yellow compounds with phenol [26,27]. The absorbance of these compounds was then measured at 490 nm at room temperature using a WFJ2100 UV-visible spectrophotometer, which was obtained from UNICO (Shanghai, China).

Accurately weighed standard glucose (10.0 mg) was placed in a 100-mL volumetric flask and water was added to the scale of 1.0, 2.0, 4.0, 8.0 and 10.0 mL, respectively, to produce a 10.0 mL solution. Glucose solution concentration gradient of 2.0 mL was taken and mixed with distilled water to 10.0 mL. Six percent phenol solution 1.0 mL was then added to it with concentrated sulfuric acid 5.0 mL, followed by shaking and then cooling. The glucose standard curve used the formula *Y* = 13.302*X* − 0.0469 (*R*^2^ = 0.9994).

##### Determination Method of Protein

This method combines Coomassie Brilliant Blue G-250 dye with a basic amino acid and protein aromatic amino acid residues in an acidic solution, which changes the position of the maximum absorption peak of the dye from 465 nm to 595 nm at room temperature, with a corresponding color change from brown-black to blue.

Accurately weighed Coomassie brilliant blue G-250 (100.0 mg) was dissolved in 95% ethanol (50.0 mL); 85% phosphoric acid (100 mL) was added, and the mixture was diluted to 1,000 mL with distilled water. Accurately weighed crystal bovine serum protein standard (10.0 mg) was diluted to 100 mL in a volumetric flask with NaCl solution (0.15 M), giving a 0.1 mg/mL albumin standard solution. 2.0, 4.0, 6.0, 8.0 and 10.0 mL of NaCl solution (0.15 M) was added to produce 10.0 mL of the solution. To 0.1 mL of the above albumin concentration gradient solution was added 5.0 mL Coomassie brilliant blue G-250 solution, shaking well. Using BSA as the standard, a protein standard curve was prepared using this Coomassie brilliant blue G-250 staining method [28]. The standard curve was drawn and used the formula *Y* = 5.865*X* + 0.0009 (*R*^2^ = 0.9994).

##### Compared Extraction Method of Polysaccharides

Dried fruit of *S. chinensis* (10.0 g) was placed in a round bottom flask and hydrothermally distilled with DES (300 mL). The mixture was refluxed for 4 h. After the extraction was complete, the supernatant was filtered, cooled, and anhydrous ethanol was slowly added to give an ethanol volume fraction of 80%. After centrifugation at 6000 rpm for 10 min with a 3K-30 ultracentrifuge (Sigma-Aldrich), the crude polysaccharide yield was calculated, and the sample was stored at −20 °C in an LGJ-10 freeze dryer, which was purchased from Brothers Equipment Co., Ltd. (Henan, China). The polysaccharide was extracted using the aqueous extraction-alcohol precipitation method, and DES was dissolved in alcohol solution and removed with liquid phase after centrifugation. Recovery of DES was done through decompression concentration.

##### Ultrasound-assisted Extraction (UE) of Polysaccharides

Dried fruit of *S. chinensis* (10.0 g) was placed in a round bottom flask and DES (300 mL) was added. Ultrasound-assisted extraction was conducted in an ultrasound/microwave extraction apparatus (XH-300A, Xianghu Sci. & Tech. Co. Ltd., Beijing, China) for 60 min at 550 W [29]. A sample was taken every 10 min to obtain the extraction kinetic curve. The same operation methods used in the section “Compared Extraction Method of Polysaccharides” were employed after the extraction was complete.

##### Microwave-assisted Extraction (ME) of Polysaccharides

Dried fruit of *S. chinensis* (10.0 g) was placed in a round bottom flask and DES (300 mL) was added. The mixture was placed in the ultrasound/microwave extraction apparatus, and microwave irradiation was conducted for 30 min at 250 W. A sample was taken every 5.0 min to obtain the extraction kinetic curve. The same operation methods used in the section “Compared Extraction Method of Polysaccharides” were employed after the extraction was complete. The extracted essential oil was dried with anhydrous sodium sulfate and stored in a freezer.

##### Alternate Ultrasound/Microwave Extraction of Polysaccharides

Dried fruit of *S. chinensis* (10.0 g) was placed in a round bottom flask and DES (300 mL) was added. The reaction mixture was extracted using alternating microwave and ultrasound treatments in the ultrasound/microwave extraction apparatus, and the extraction procedure was optimized. According to Table 3, the experiments A-G used total polysaccharide extraction time of 60 min. The extraction process was optimized by changing the extraction time and extraction sequence of the ultrasound/microwave under the same conditions, and using a reflux extraction method (experiment H) for comparison. The same operation methods used in the section “Compared Extraction Method of Polysaccharides” were employed after the extraction was complete.

##### RSM Optimization Parameters of Polysaccharides

To further study the interaction between the factors, we optimized the operating conditions by RSM and used the Box-Behnken software for data processing [29]. The Box-Benhnken design with three factors was applied using Design-Expert 8.0 without any blocking. The bounds of the ultrasound factors were 1:20–1:40 solid–liquid ratio, 400–700 W ultrasound power, 30–50 min extraction time. Specific protocols for experimental conditions are shown in Table 7.

Similarly, the bounds of the microwave factors were 1:20–1:40 solid-liquid ratio, 200–250 W microwave power, 15–25 min extraction time. Specific protocols for experimental conditions are shown in Table 8.

#### 3.3.2. Sevag Deproteinization Method for Polysaccharides

For preparation of 2.0% (mass fraction) crude polysaccharide solution, a volume of 1/3 chloroform-*n*-butanol (4:1 *v*/*v*) solution was added, full oscillated for 15 min, oscillated twice, centrifuged to remove the precipitate. A small amount of water was taken to measure protein content and polysaccharide content, then the aqueous phase was added to the 1/3 chloroform-*n*-butanol solution and repeated twelve times. The deproteinized polysaccharide solution was concentrated and lyophilized at −20 °C [30], and the protein removal efficiency and reserved amount of polysaccharide were calculated.

#### 3.3.3. Hydroxyl Free Radical Scavenging Method for Polysaccharides

*S. chinensis* polysaccharide (0.05–0.50 mg/mL) was added to a mixed solution (3 mL) containing 2-deoxyribose (3 mM), ferric chloride (0.1 mM), hydrogen peroxide (1 mM), ethylenediaminetetraacetic acid (0.1 mM), VC (0.1 mM), and phosphate buffer solution (0.02 M, pH 7.4), and heated at 37 °C for 1 h. Next, tert-Butanol (1 mL, 1%) and trichloroacetic acid (1 mL, 2.8%) solutions were added, followed by heating at 100 °C for 20 min. The reaction was then cooled to room temperature and analyzed at 536 nm. The ability of *S. chinensis* polysaccharide to scavenge hydroxyl free radical with polysaccharide concentration was analyzed. The unit of polysaccharide concentration was mg/mL, and the unit of cell proliferation rate was %.

#### 3.3.4. In vitro MTT Activity Test Method for Polysaccharides

Polysaccharides are a class of active ingredients that can regulate immune activity and delay aging [31]. The amount of crystals produced was proportional to the number of living cells (succinate dehydrogenase in dead cells disappeared and MTT cannot be reduced). The mouse splenic lymphocytes were cultured in DMEM with 10% FBS. First, cell suspensions were seeded in 96-well plates (1 × 105/well), and incubated at 37 °C for 12 h; then the polysaccharides were added. After 6 h, a certain amount of MTT were added into each well and the plate was further incubated for 4 h. Finally, the medium was removed and 200 μL DMSO was added to dissolve the formazan. After 10 min, absorbance at 490 nm was measured [32].

#### 3.3.5. Essential Oil Extraction Methods

##### Hydro-distillation (HD) Extraction of Essential Oils

Dried fruit (10.0 g) was hydrodistilled using a Clevenger-type instrument and extracted with DES (300 mL) for 4 h until no further essential oil was obtained. The essential oil was collected, dried over anhydrous sodium sulfate, and stored at 0 °C [33].

##### Microwave-assisted Hydro-distillation Extraction (MHD) of Essential Oils

The MHD process is based on a conventional hydro-distillation system, but with microwave irradiation used during the heating process. An ultrasound/microwave extraction apparatus was used to heat the sample, a mixture of *S. chinensis* (10.0 g) and DES (300 mL), in which the microwave power was 250 W.

##### Ultrasound/Microwave-assisted Hydro-distillation (UMHD) Extraction of Essential Oils

Dried fruit (10.0 g) were mixed with DES (300 mL, 0.75 M) in a round bottom flask. The ultrasound power was 550 W, and the microwave power was 250 W. After UMHD extraction, the essential oils were collected.

#### 3.3.6. Chemical Composition Analysis of Essential Oil

GC-MS analysis of the essential oil was carried out on an Agilent 6890N-5973 insert gas chromatograph (Agilent Technologies, Palo Alto, CA, USA) with a capillary column (30mm × 0.25 mm, film thickness 0.25 μm) equipped with an Agilent 6890N-5973 mass selective detector in the electron impact mode. The GC was operated under the following conditions: manual injection 1 μL; injector temperature, 270 °C; carrier gas (He) flow 1 mL/min; oven temperature programmed 40 °C to 165 °C at a rate of 10 °C/min, from 165 °C to 200 °C at a rate of 5 °C/min and 200 °C to 250 °C at a rate of 10 °C/min. The detector temperature was 280 °C. The MS was operated under 70 eV, scan range 15–500 amu, scan-TIC. The chemical compositions of essential oils were identified by direct comparison of their mass spectral pattern in NIST02 Mass Spectral Library.

#### 3.3.7. Scanning Electron Microscopy (SEM)

Microstructure analysis was performed using a scanning electron microscope (FEI QUANTA200, Eindhoven, Netherlands). To obtain conductivity, a thin layer of gold was sputtered onto the sample surface using an SCD 005 Sputter Coater (BAL-TEC, Aavorstadt, Switzerland).

#### 3.3.8. Antioxidant Analysis Methods for Essential Oils

##### Free Radical Scavenging Activity

A 1.0 mL methanolic solution of essential oil was mixed with 2.0 mL methanolic solution of DPPH (35 mg/L). The absorbance of mixture was measured at 517 nm after 30 min in darkness. VC was used as the positive control [34,35].

##### Ferric Reducing/Antioxidant Power (FRAP) Assay

The capacity for reducing antioxidants was determined using the FRAP assay [36]. Tests were performed in triplicate and repeated with VC as the control. The reducing antioxidant power of the sample was expressed in Trolox equivalents, which is the ratio of the slope of the sample regression line to the Trolox slope.

##### Scavenging efficiency of β-carotene

A stock solution of *β*-carotene-linoleic acid was prepared as follows [37,38]. The procedure was repeated with the same concentration of VC and a blank containing only ethanol (500 µL). After incubation, the absorbance of the mixture was measured at 490 nm.

## 4. Conclusions

DES was found as a green medium to successfully extract high purity *S. chinensis* polysaccharides and essential oils. The optimal reagent was prepared from choline chloride and ethylene glycol with molar ratio 1:3 and solvent ratio 7:3 *v*/*v*. By optimizing the ultrasonic/microwave, an extraction condition of 550W (UE), 250W (ME), and solid-liquid ratio of 1:30 *w*/*v* was obtained. Besides, in the experiment of *S. chinensis* polysaccharides, it was found that the crude polysaccharide was deproteinized four times by Sevag method to obtain maximum fine polysaccharide (5.11 g/100g); in the scavenging activity of *S. chinensis* polysaccharide toward hydroxyl radical, we found when the sampling concentration was more than 0.20 mg/mL, and the scavenging activity had almost doubled. The in vitro activity test of polysaccharides showed that 0.20 mg/mL was the optimal concentration of *S. chinensis* polysaccharides with Lps cells—it was about 1.5-fold better than that with ConA cells. Similarly, the essential oil extraction method was compared and 400 W (UE) and 20 min were selected. In the antioxidant experiments of essential oils, the free radical scavenging activity of DES UMHD extracts determined by β-carotene is first-rate. Furthermore, the FRAP assay of essential oils showed that the DES UMHD sample had optimal antioxidant activity. This indicates that DES presents an efficient separation of essential oils and polysaccharides from natural plants under ultrasound/microwave digestion.

## Figures and Tables

**Figure 1 molecules-24-01288-f001:**
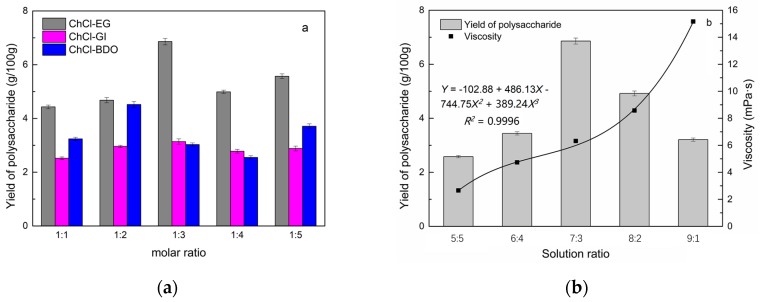
Screening of DES: (**a**) Effect of DES composition on the extraction yield of polysaccharide, (**b**) effect of water amount in the DES solution on the extraction yield of polysaccharide.

**Figure 2 molecules-24-01288-f002:**
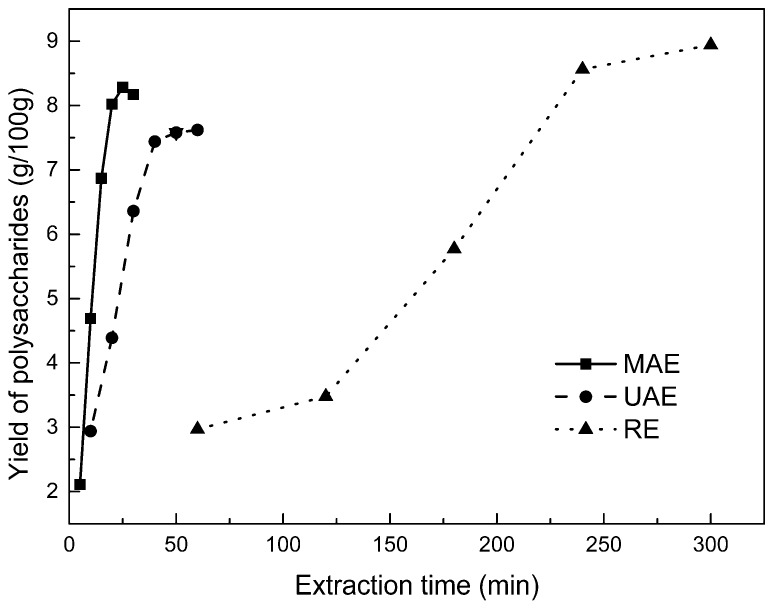
Kinetic study of polysaccharide extraction.

**Figure 3 molecules-24-01288-f003:**
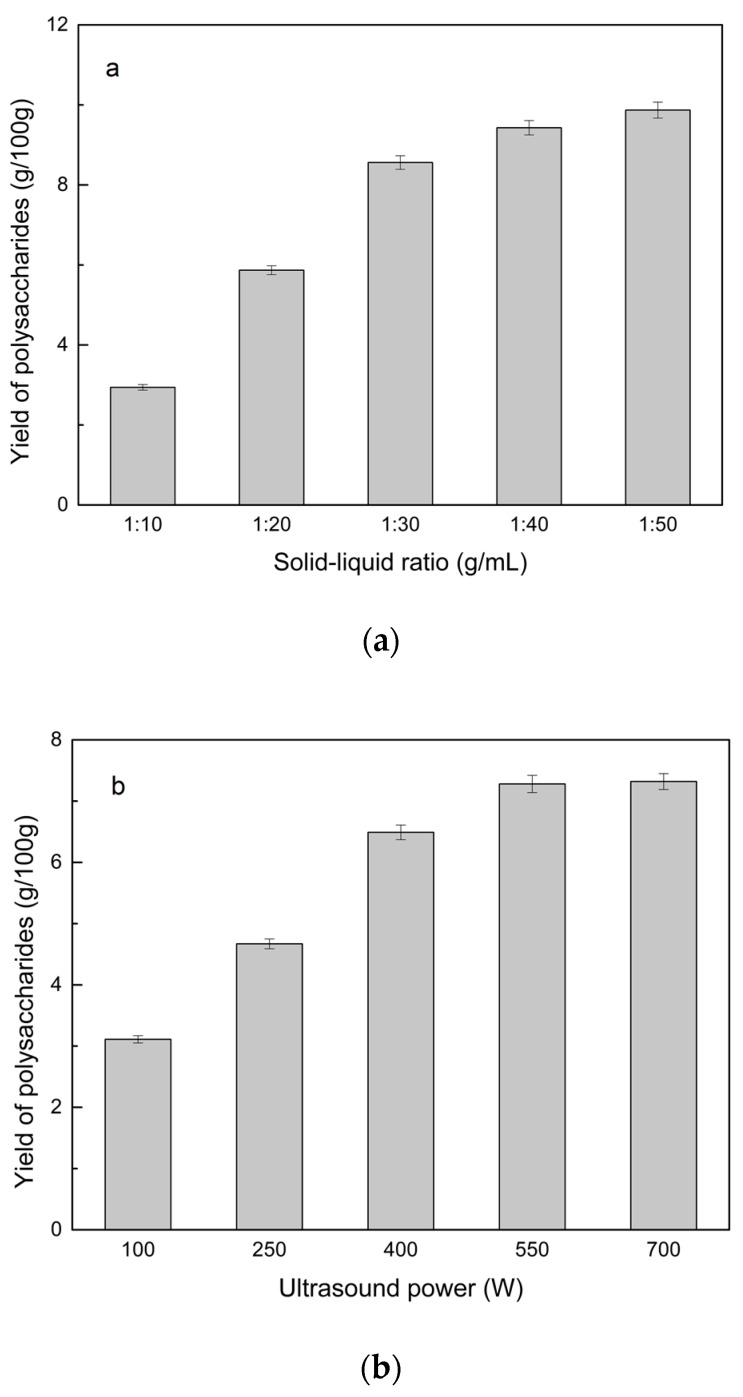
Single factor analysis of polysaccharide extraction. Effect of extraction parameters on polysaccharide yield: (**a**) Solid-liquid ratio, (**b**) ultrasound power, and (**c**) microwave power.

**Figure 4 molecules-24-01288-f004:**
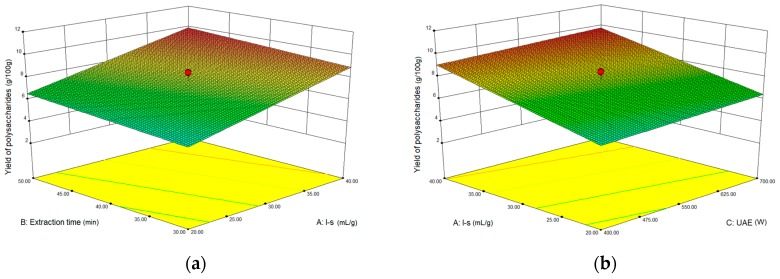
The response surface for the effect of independent variables on ultrasound-assisted extraction efficiency of polysaccharides. (**a**) Solid–liquid ratio and extraction time, (**b**) solid–liquid ratio and ultrasound power.

**Figure 5 molecules-24-01288-f005:**
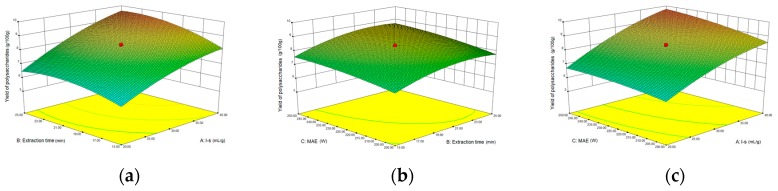
The response surface for the effect of independent variables on microwave-assisted extraction efficiency of polysaccharides. (**a**) Solid–liquid ratio and extraction time, (**b**) extraction time and microwave power, (**c**) solid–liquid ratio and microwave power.

**Figure 6 molecules-24-01288-f006:**
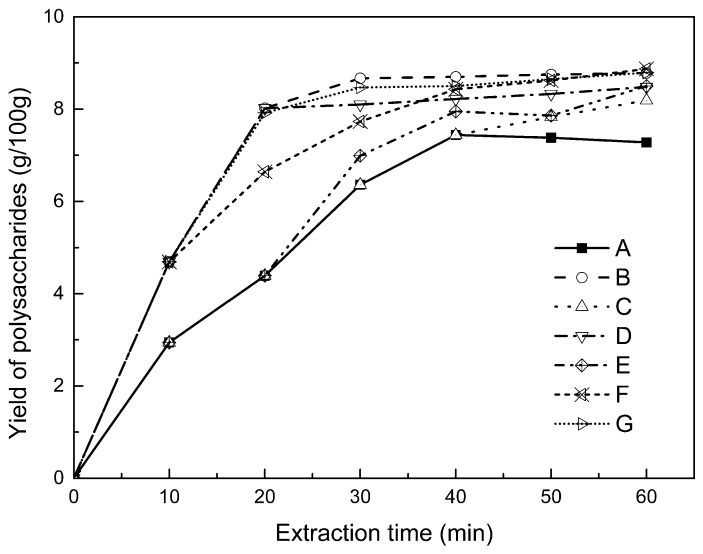
Alternate and synergistic ultrasound/microwave extraction of polysaccharides.

**Figure 7 molecules-24-01288-f007:**
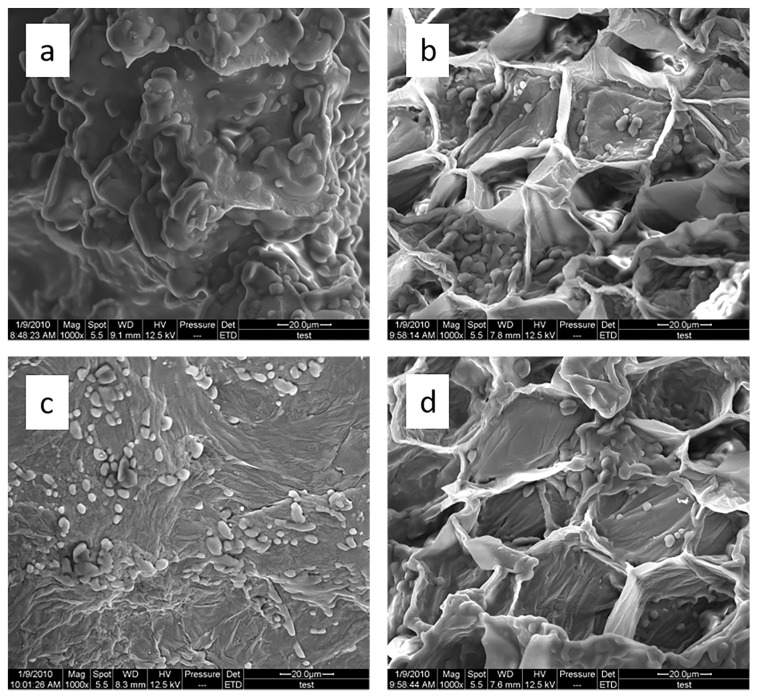
SEM images of *S. chinensis* fruits. (**a**) *S. chinensis* fruit raw materials, (**b**) *S. chinensis* fruits treated by ultrasound digestion, (**c**) *S. chinensis* fruits treated by microwave digestion, and (**d**) *S. chinensis* fruits treated by alternate ultrasound/microwave digestion.

**Figure 8 molecules-24-01288-f008:**
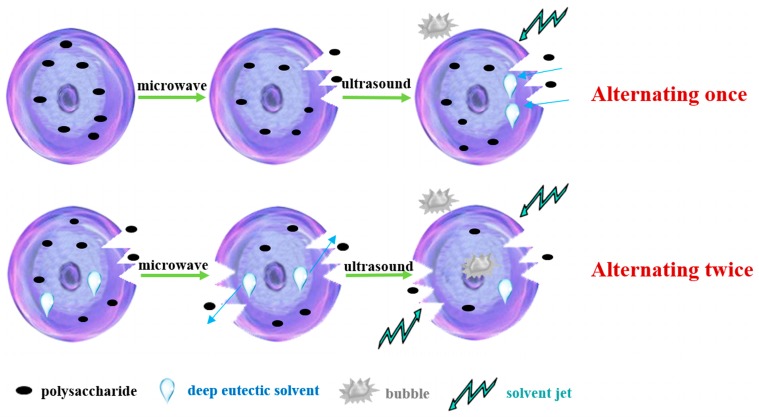
The mechanism of alternating ultrasound and microwave-assisted extraction.

**Figure 9 molecules-24-01288-f009:**
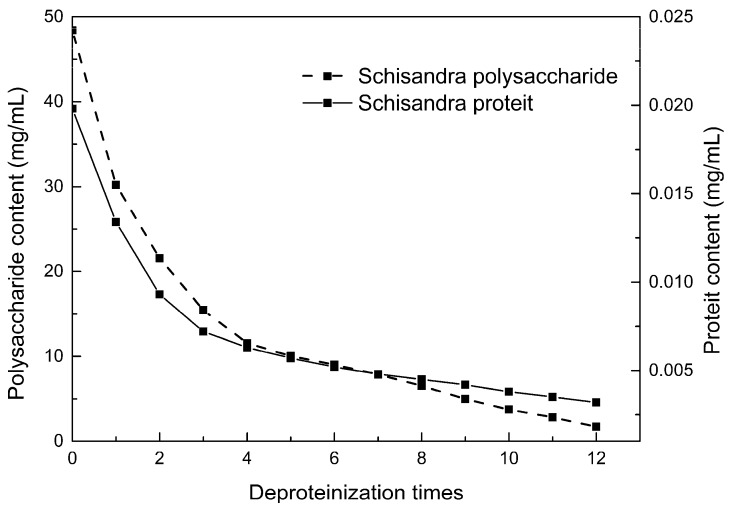
Polysaccharide content after removing protein 12 times with the Sevag method.

**Figure 10 molecules-24-01288-f010:**
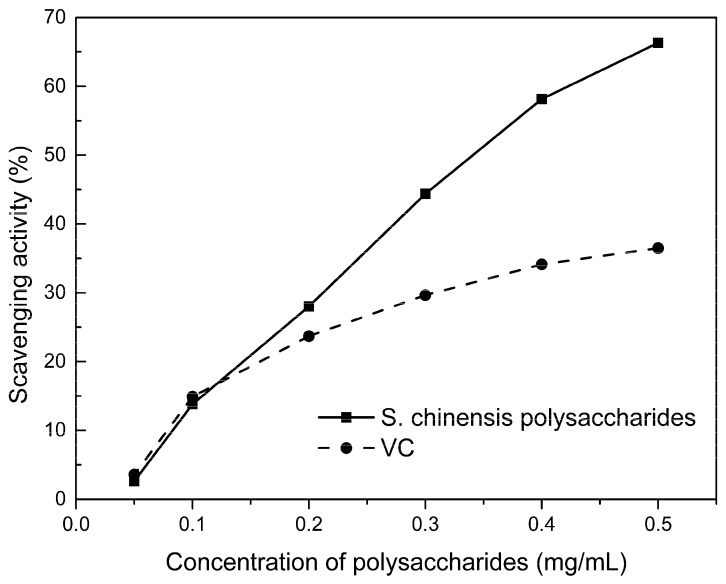
Hydroxyl radical scavenging activity of polysaccharide.

**Figure 11 molecules-24-01288-f011:**
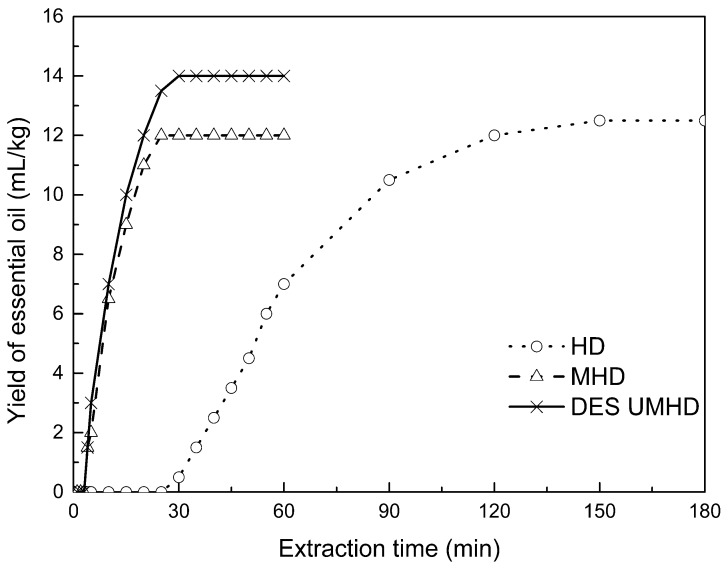
Kinetic study of essential oil extraction.

**Figure 12 molecules-24-01288-f012:**
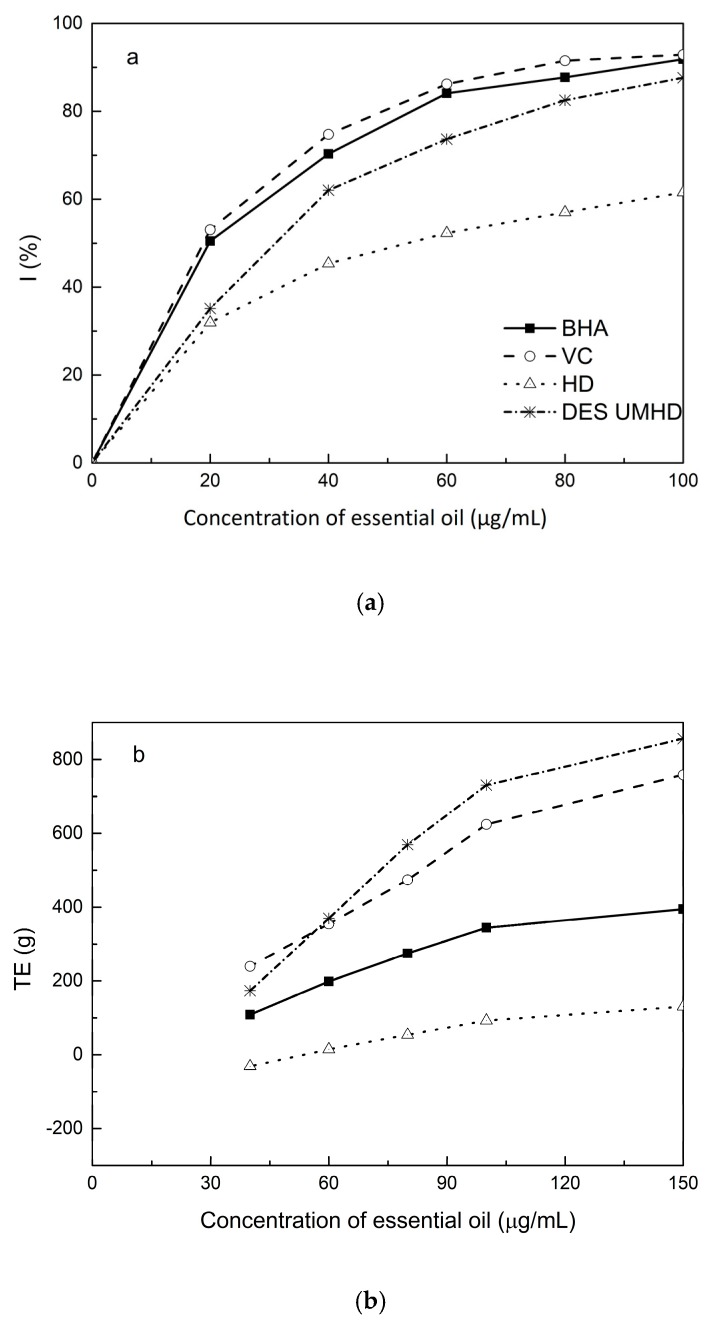
Antioxidant abilities of essential oils: (**a**) DPPH free radical scavenging activity of essential oils, (**b**) FRAP ferric-reducing antioxidant power of essential oils, and (**c**) β-carotene scavenging capacity of essential oils.

**Figure 13 molecules-24-01288-f013:**
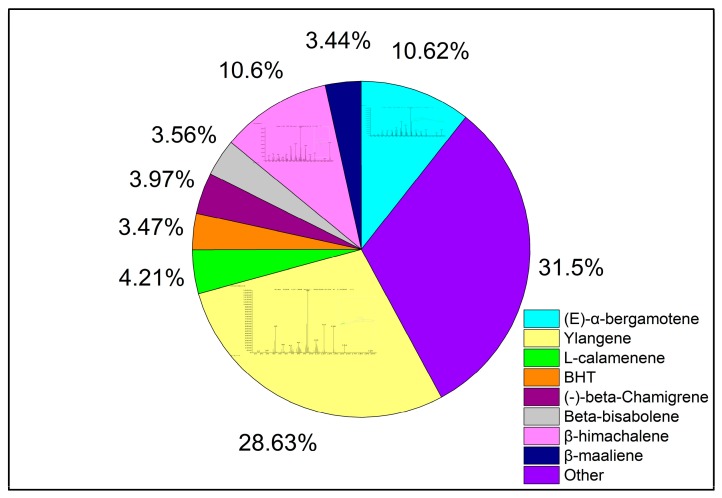
Volatile compounds in *S. chinensis* obtained by DES UMHD.

**Table 1 molecules-24-01288-t001:** Test of significance for regression coefficient for ultrasound-assisted extraction.

Source	Sum of Squares	df	Mean Square	F-Value	p-Value Prob > F
Model	43.18	3	14.39	19.51	<0.0001
A:liquid-solid (mL/g)	38.00	1	38.00	51.50	<0.0001
B:Extraction time (min)	3.31	1	3.31	4.48	0.0502
C:UAE power (W)	1.87	1	1.87	2.53	0.1310
Residual	11.81	16	0.74	——	——
Lack of Fit	11.78	11	1.07	247.97	<0.0001
Pure Error	0.022	5	4.320 × 10^−3^	——	——
Cor Total	54.98	19	——	——	——

**Table 2 molecules-24-01288-t002:** Test of significance for regression coefficient for microwave-assisted extraction.

Source	Sum of Squares	df	Mean Square	F-Value	p-Value Prob > F
Model	34.91	9	3.88	17.30	<0.0001
A:liquid-solid (mL/g)	24.17	1	24.17	107.82	<0.0001
B:Extraction time (min)	3.45	1	3.45	15.39	0.0029
C:MAE power (W)	2.27	1	2.27	10.14	0.0098
AB	0.42	1	0.42	1.87	0.2017
AC	0.26	1	0.26	1.17	0.3043
BC	0.023	1	0.023	0.10	0.7548
A^2^	1.73	1	1.73	7.70	0.0196
B^2^	2.90	1	2.90	12.93	0.0049
C^2^	0.33	1	0.33	1.48	0.2523
Residual	2.24	10	0.22	——	——
Lack of Fit	2.20	5	0.44	47.86	0.0003
Pure Error	0.046	5	9.177 × 10^−3^	——	——
Cor Total	37.16	19	——	——	——

**Table 3 molecules-24-01288-t003:** Experimental program of alternating ultrasound/microwave extractions of polysaccharides.

Item	Experimental Program	Essential Oil Yield (mL/kg)	Polysaccharide Yield (g/100g)	Environmental Impact (g CO_2_ Rejected)
A	UAE 60 min	10.6	7.02 ± 0.11	440
B	MAE 60 min	10.9	7.83 ± 0.12	200
C	UAE 40 min + MAE 20 min	11.2	8.19 ± 0.20	360
D	MAE 20 min + UAE 40 min	11.5	8.48 ± 0.33	360
E	UAE 20 min + MAE 10 min + UAE 20 min + MAE 10 min	11.6	8.50 ± 0.31	360
F	MAE 10 min + UAE 20 min + MAE 10 min + UAE 20 min	12.2	8.87 ± 0.29	360
G	UAE & MAE 60 min	11.9	8.79 ± 0.26	640
H	Reflux extraction 4 h	11.7	8.56 ± 0.30	3200

**Table 4 molecules-24-01288-t004:** Effect of polysaccharide concentration on mouse spleen cell proliferation.

Concentration	Parallel Sample 1		Parallel Sample 2		Parallel Sample 3	
(mg/mL)	PR/%	A	PR/%	A	PR/%	A
0.80	1.54	0.7268	2.53	0.7339	4.62	0.7489
0.40	24.35	0.8914	24.36	0.8902	25.44	0.8979
0.20	29.48	0.9268	30.58	0.9347	29.30	0.9255
0.10	29.66	0.9281	23.54	0.8843	26.15	0.9030
0.05	20.86	0.8651	22.63	0.8778	23.40	0.8833
Control group	——	0.7194	——	0.7118	——	0.7162

**Table 5 molecules-24-01288-t005:** Effect of adding ConA and Lps on mouse spleen cell proliferation.

Concentration	Parallel Sample 1		Parallel Sample 2		Parallel Sample 3	
(mg/mL)	PR/%	A	PR/%	A	PR/%	A
0.20	29.48	0.9268	30.58	0.9347	29.30	0.9255
0.20 + ConA	108.76	1.4943	107.11	1.4825	108.97	1.4958
0.20 + Lps	146.76	1.7663	143.15	1.7405	150.42	1.7925
ConA	79.04	1.2816	80.33	1.2908	80.53	1.2922
Lps	79.06	1.2817	79.23	1.2829	78.85	1.2802
Control group	——	0.7194	——	0.7118	——	0.7162

**Table 6 molecules-24-01288-t006:** Volatile compounds in *S. chinensis* essential oil obtained by HD, MHD and DES UMHD.

No.	Retention Time (min)	Compounds	Alias Name	CAS Number	Molecular Formular	RA %		
						DES UMHD	MHD	HD
1	4.94	*α*-Pinene	α-Pinene	7785-70-8	C_10_H_16_	0.82	——	——
2	5.68	Camphene	Camphene	79-92-5	C_10_H_16_	0.78	——	——
3	8.02	Bicyclo[4.1.0]hept-2-ene, 3,7,7-trimethyl	2-Carene	554-61-0	C_10_H_16_	0.61	——	——
4	8.40	D-Limonene	D-Limonene	5989-27-5	C_10_H_16_	0.68	——	——
5	9.22	Benzene, 1-methyl-2-(1-methylethyl)	*o*-Cymene	527-84-4	C_10_H_14_	0.39	——	——
6	9.61	1,4-Cyclohexadiene, 1-methyl-4-(1-methylethyl)	gamma-Terpinene	99-85-4	C_10_H_16_	0.34	——	——
7	16.43	Benzene, 2-methoxy-4-methyl-1-(1-methylethyl)	Thymol methyl ether	1076-56-8	C_11_H_16_O	0.79	1.26	1.13
8	17.79	Bicyclo[2.2.1]heptan-2-ol, 1,7,7-trimethyl-, acetate	L-Bornyl acetate	5655-61-8	C_12_H_20_O_2_	1.53	2.04	2.18
9	18.22	Ylangene	Ylangene	14912-44-8	C_15_H_24_	28.63	38.57	37.60
10	19.16	Cyclobuta[1,2:3,4]dicyclopentene, decahydro-3a-methyl-6-methylene-1-(1-methylethyl)	β-bourbonene	5208-59-3	C_15_H_24_	0.49	——	——
11	19.55	Cyclohexane, 1-ethenyl-1-methyl-2,4-bis(1-methylethenyl)	(-)-β-Elemene	515-13-9	C_15_H_24_	0.72	——	——
12	21.10	1,6-Cyclodecadiene, 1-methyl-5-methylene-8-(1-methylethyl)	Germacrene D	23986-74-5	C_15_H_24_	0.54	——	——
13	22.81	1,6,10-Dodecatriene, 7,11-dimethyl-3-methylene	(E)-(β)-Farnesene	18794-84-8	C_15_H_24_	1.34	——	——
14	25.00	Bicyclo[3.1.1]hept-2-ene, 2,6-dimethyl-6-(4-methyl-3-pentenyl)	(E)-α-bergamotene	13474-59-4	C_15_H_24_	10.62	11.20	10.57
15	25.32	Cyclohexane, 1-ethenyl-1-methyl-2-(1-methylethenyl)-4-(1-methylethylidene)	Elixene	3242-08-8	C_15_H_24_	1.13	——	——
16	25.64	Naphthalene, 1,2,4a,5,6,8a-hexahydro-4,7-dimethyl-1-(1-methylethyl)	α-amorphene	483-75-0	C_15_H_24_	1.88	2.55	2.51
17	26.11	Spiro[5.5]undec-2-ene, 3,7,7-trimethyl-11-methylene	(-)-β-Chamigrene	18431-82-8	C_15_H_24_	3.97	4.33	5.19
18	26.35	Cyclohexene, 1-methyl-4-(5-methyl-1-methylene-4-hexenyl)	β-bisabolene	495-61-4	C_15_H_24_	3.56	——	——
19	26.47	1H-Cyclopropa[a]naphthalene, 1a,2,3,3a,4,5,6,7b-octahydro-1,1,3a,7-tetramethyl	β-maaliene	489-29-2	C_15_H_24_	3.44	2.45	3.55
20	27.07	1H-Benzocycloheptene, 2,4a,5,6,7,8-hexahydro-3,5,5,9-tetramethyl	β-himachalene	1461-03-6	C_15_H_24_	10.6	11.88	10.54
21	27.34	Spiro[5.5]undeca-1,8-diene, 1,5,5,9-tetramethyl	a-Chamigrene	19912-83-5	C_15_H_24_	1.27	1.13	1.18
22	27.74	Tricyclo[5.4.0.0(2,8)]undec-9-ene, 2,6,6,9-tetramethyl	(+)-α-Longipinene	5989-08-2	C_15_H_24_	3.05	3.10	3.12
23	28.25	Isoledene		95910-36-4	C_15_H_24_	0.74	1.04	0.96
24	28.72	Benzene, 1-methyl-4-(1,2,2-trimethylcyclopentyl)	(+)-Cuparene	16982-00-6	C_15_H_22_	1.37	1.75	1.95
25	29.85	Bicyclo[7.2.0]undec-4-ene, 4,11,11-trimethyl-8-methylene	b-Caryophyllene	87-44-5	C_15_H_24_	0.66	——	——
26	30.58	Naphthalene, 1,2-dihydro-1,1,6-trimethyl		55682-80-9	C_15_H_16_	0.47	——	——
27	30.68	alpha-Calacorene		21391-99-1	C_15_H_20_	0.23	0.24	0.38
28	30.86	Azulene, 1,2,3,4,5,6,7,8-octahydro-1,4-dimethyl-7-(1-methylethylidene)	Guaiene	88-84-6.	C_15_H_24_	0.55	0.40	0.55
29	33.63	Benzenemethanol, 2,4-dimethyl		16308-92-2	C_9_H_12_O	3.02	——	——
30	33.80	Tricyclo[4.4.0.0(2,7)]dec-3-ene-3-methanol, 1-methyl-8-(1-methylethyl)		41610-69-9	C_15_H_24_O	3.47	3.28	3.8
31	34.66	1H-Cycloprop[e]azulene, 1a,2,3,5,6,7,7a,7b-octahydro-1,1,4,7-tetramethyl	(+)-LEDENE	21747-46-6	C_15_H_24_	0.69	——	——
32	36.45	Naphthalene, 1,2,3,4-tetrahydro-1,6-dimethyl-4-(1-methylethyl)	L-calamenene	6617-49-8	C_15_H_22_	4.21	5.69	5.86
33	37.79	Cyclohexane, 1,2-diethenyl-4-(1-methylethylidene)			C_13_H_20_	0.76	——	——
34	38.50	4,6,6-Trimethyl-2-(3-methylbuta-1,3-dienyl)-3-oxatricyclo[5.1.0.0(2,4)]octane			C_15_H_22_O	3.34	5.22	5.08
35	39.62	Benzenemethanol, 3,5-dimethyl		27129-87-9	C_9_H_12_O	2.07	3.84	3.84
36	43.10	Dibutyl phthalate		84-74-2	C_16_H_22_O_4_	1.19	——	——
Total						99.95	99.97	99.99

**Table 7 molecules-24-01288-t007:** Experimental design matrix to screen important variables for ultrasound-assisted extraction.

Run	Factor A	Factor B	Factor C	Response
	L–S (mL/g)	Extraction Time (min)	UAE (W)	Yield of Polysaccharides (g/100g)
1	40.00	50.00	400.00	9.39
2	40.00	50.00	700.00	9.45
3	40.00	30.00	400.00	9.11
4	46.82	40.00	550.00	9.77
5	30.00	40.00	297.73	5.59
6	20.00	30.00	700.00	6.09
7	40.00	30.00	700.00	9.26
8	30.00	40.00	550.00	8.47
9	30.00	56.82	550.00	8.74
10	20.00	50.00	400.00	6.82
11	30.00	40.00	550.00	8.52
12	30.00	40.00	550.00	8.56
13	20.00	50.00	700.00	6.22
14	13.18	40.00	550.00	3.55
15	30.00	40.00	550.00	8.44
16	20.00	30.00	400.00	5.76
17	30.00	23.18	550.00	5.73
18	30.00	40.00	550.00	8.37
19	30.00	40.00	550.00	8.46
20	30.00	40.00	802.27	8.63

**Table 8 molecules-24-01288-t008:** Experimental design matrix to screen important variables for microwave-assisted extraction.

Run	Factor A	Factor B	Factor C	Response
	l-s (mL/g)	Extraction Time (min)	MAE (W)	Yield of Polysaccharides (g/100g)
1	40.00	15.00	200.00	7.33
2	40.00	25.00	250.00	9.56
3	46.82	20.00	225.00	10.11
4	30.00	20.00	225.00	8.34
5	30.00	20.00	225.00	8.43
6	20.00	25.00	250.00	6.84
7	20.00	15.00	250.00	6.33
8	30.00	20.00	225.00	8.21
9	30.00	20.00	225.00	8.44
10	30.00	20.00	225.00	8.24
11	20.00	25.00	200.00	6.02
12	13.18	20.00	225.00	4.45
13	40.00	15.00	250.00	8.66
14	30.00	28.41	225.00	8.32
15	30.00	20.00	267.04	8.52
16	30.00	20.00	225.00	8.37
17	20.00	15.00	200.00	6.25
18	30.00	11.59	225.00	5.66
19	40.00	25.00	200.00	8.54
20	30.00	20.00	182.96	7.14

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
