# Peer review of "Alternate Ultrasound/Microwave Digestion for Deep Eutectic Hydro-distillation Extraction of Essential Oil and Polysaccharide from Schisandra chinensis (Turcz.) Baill"

_molecules, 2019, doi:10.3390/molecules24071288_

Round 1
Reviewer 1 Report
Please indicate what is ED, GI and BDO when mentioned for the first time (line 89).
Yield of polysaccharide is expressed as g/100 g. Is it 100 g of dry plant or fresh plant material?
A sentence "Results showed that the DES solvent extracted chitin showed a high yield with a maximum range of 94 68.6±1.0 g/kg for ChCl-EG 1:3." makes no sense. Furthermore, why is it expressed in g/kg, while in Fig.1 you have g/100 g?
the authors should explain what is VC when mentioned for the first time (it is explained in materials and methods, which comes after the discussion section).
The authors should explain why the hydroxyl radical scavenging activity was better at higher concentrations of polysaccharides compared to VC?
I am not quite convinced that you can remove DES with freeze-drying, I am sure that only water is removed, and DES components remain in your sample. You should establish a method to prove there are no DES components in your sample. So, if DES components remain in your sample, they will influence all the activities you investigated, as well as the extraction yield and so on.
When the authors explain the hydro-distillation using DES they do mention a temperature, which is very important when you use DESs, since they tend to decompose at high temperatures.
Furthermore, it is well known that DES (ChCl:EG) is polar, so the authors should explain how they managed to extract slightly polar or non polar components with a polar solvent.
The authors claim they determined the volatile compounds in their extracts, but ther is no methodology on how they performed it? If GC/MS was used, it should be explained.
Author Response
Please indicate what is ED, GI and BDO when mentioned for the first time (line 89).
Response: I’m sorry for my mistake, and the definitions of ED, GI and BDO were added in Section 2.1.
Yield of polysaccharide is expressed as g/100 g. Is it 100 g of dry plant or fresh plant material?
Response: The response value in the whole text, including Tables and Figures has been revised to the yield of polysaccharide expressed as g/100g.
A sentence "Results showed that the DES solvent extracted chitin showed a high yield with a maximum range of 94 68.6±1.0 g/kg for ChCl-EG 1:3." makes no sense. Furthermore, why is it expressed in g/kg, while in Fig.1 you have g/100 g?
Response: The sentence was revised to “Results showed that the DES solvent (ChCl-EG 1:3 molar ratio) extracted chitin showed a high yield of 6.86±0.10 g/100g” in section 2.1. The response value in the whole text, including Tables and Figures has been revised to the yield of polysaccharide expressed as g/100g.
The authors should explain what is VC when mentioned for the first time (it is explained in materials and methods, which comes after the discussion section).
Response: VC is "vitamin C" and was added in Section 2.5.
The authors should explain why the hydroxyl radical scavenging activity was better at higher concentrations of polysaccharides compared to VC?
Response: According to Fenton reaction, there were three reagents, oxidant, reducer and metal ion, and the generation of a hydroxyl radical depends on the content of Fe2+ and H2O2. However, there were several nature metal ions (such as Fe, Zn, Cu) in polysaccharide. Moreover, if the absolute concentrations of Fe2+ higher than H2O2 and VC, a part of VC was used to reduce Fe3+, which results in inhibition of H2O2. Finally, the Fenton reaction was inhibited and •OH was not generated. Therefore, the hydroxyl radical scavenging activity was better at higher concentrations of polysaccharides compared to VC, and the reason was also added in Section 2.5.
I am not quite convinced that you can remove DES with freeze-drying, I am sure that only water is removed, and DES components remain in your sample. You should establish a method to prove there are no DES components in your sample. So, if DES components remain in your sample, they will influence all the activities you investigated, as well as the extraction yield and so on.
Response: The polysaccharide was extracted by aqueous extraction-alcohol precipitation method, and during precipitation process, the DES was dissolved in alcohol solution and removed with liquid phase after 6000 r/min centrifugation, and then the polysaccharide was dried with freeze-drying.
When the authors explain the hydro-distillation using DES they do mention a temperature, which is very important when you use DESs, since they tend to decompose at high temperatures.
Response: The traditional hydro-distillation method was used to extract essential oil, the essential oil was spill over with the boiling water at 100ºC, and The polysaccharide was extracted by aqueous extraction-alcohol precipitation method. Thus, the DES hydro-distillation method was suitable for simultaneously extracting crude polysaccharides and essential oils, and the crude polysaccharide was precipitated by subsequently increasing the alcohol content of the DES solution, and then filtered, which completely separated polysaccharide from DES solution.” in Section 2.7.
Furthermore, it is well known that DES (ChCl:EG) is polar, so the authors should explain how they managed to extract slightly polar or non polar components with a polar solvent.
Response: Polysaccharides were extracted according to the “Similar Dissolves Similar” Theory, so the polysaccharides has the similar polarity with DES; while the essential oil was based on the Daltons law of partial pressures, using conventional hydro-distillation method, and the essential oil was spill over with the boiling water at 100ºC.
The authors claim they determined the volatile compounds in their extracts, but ther is no methodology on how they performed it? If GC/MS was used, it should be explained.
Response: GC-MS analysis of the essential oil was carried out on an Agilent 6890N-5973 insert gas chromatograph (Agilent Technologies, Palo Alto, CA, USA) with a capillary column (30mm × 0.25mm, film thickness 0.25 μm) equipped with an Agilent 6890N-5973 mass selective detector in the electron impact mode. The GC was operated under the following conditions: manual injection 1 μl; injector temperature, 270 °C; carrier gas (He) flow 1 ml/min; oven temperature programmed 40 °C to 165 °C at a rate of 10 °C /min, from 165 °C to 200 °C at a rate of 5 °C/min and 200 °C to 250 °C at a rate of 10 °C/min. The detector temperature was 280 °C. The MS was operated under 70 eV, scan range 15–500 amu, scan-TIC. The chemical compositions of essential oils were identified by direct comparison of their mass spectral pattern in NIST02 Mass Spectral Library. This method was added in section 3.3.6.
Reviewer 2 Report
Use DES in green extraction process is a novelty and interesting option, in addition, S. chinensis is a promising source of bioactive substances. However, the article requires a wide revision before to be considered for publication. Below more details:
Introduction
- Lines 36-37: Cite more specific reports about uses of DESs for recovery of bioactive compounds from natural sources, particularly antioxidant substances. ¿There is reports on S. chinensis or similar species (e.g. same genere)?
- Lines 58-60: The authors cited several properties of S. chinensis, however, only present one reference. In addition, the article cited by the authors (ref number [16]) is a review about antibacterial properties of essential oils from medicinal plants. The authors should present more references relative to the biological properties of S. chinensis.
- Lines 70-12: Present the suitable references
- Lines 61-65: ¿What is the relationship between these sentences and the bioactive properties of S. chinensis?, I do not see it, please improve this paragraph.
- Lines 78-81: Move to results and discussion section
Materials and Methods
- Provide more details about sampling process; e.g. maturity of S. chinensis fruits, ¿how much sample was sampling? ¿the sampling was aleatory? etc
- line 325: ¿how much ChCl and EG were used?
- line ¿How much?
- Line 337: This information is not necessary
- Line 330: I suggest change the section title, e.g. "Polysaccharide extraction/recovery and analysis"
- Line 330: I suggest change the section title, e.g. "Polysaccharide extraction/recovery and analysis". Additionally, present first the extraction procedure (sections 3.3.1.3 and 3.3.1.4), and after the analysis procedure (sections 3.3.1.1 and 3.3.1.2).
- Sections 3.3.1.1 and 3.3.1.2: More specific details about extracts analysis should be cited: e.g. ¿Did you use any preparation procedure for the extracts before analysis them? ¿what amount or concentration of extracts was used? ¿what units used to present the results?
- Lines 346-347; 355-356; 360-361: ¿temperature?
- The section 3.3.1.6 is not clear: ¿How the alternating treatments were applied? ¿How the extraction process was optimized? ¿experimental design?
- Section 3.3.3 ¿what units used to present the results?
- Section 3.3.4 is not clear
- Section 3.3.5: ¿temperature?
Results and discussion
- The molar ratios explored were not cite in methods
- Line 95: ¿How this optimal condition was established? ¿Experimental design?
- Lines 95-98: ¿These results are important? ¿why?
- After extraction process, ¿how the DESs were recovered?
- Section 2.2.1: Kinetic study was not described in the methods section
- Line 111: ¿optimal extraction times? I suggest not use the word "optimal". In this work the authors did not optimize the extraction process.
- Section 2.2.1 and 2.2.2: The experimental design to optimize the extraction process is not clear. For RE the authors “optimized” each variable independently, however they not consider the interactions between variables (e.g. time interaction with solid-liquid ratio)
- Section 2.2.3: ¿What is the interaction between the extraction time and the power?
- Sections 2.2: Is necessary discuss results
- Section 2.10 should be move at section in order to explain the alternate UA/M extraction process
- Figure 6: ¿What polysaccharides extract was used?
- Explain how the polysaccharides scavenging the OH* radical
- Section 2.7: similar to the comments of section 2.2.1
- Section 2.8: results requires discussion
- Section 2.9: ¿What is the relationship between chemical composition and the observed antioxidant properties? The Chemical composition should be related with the biological properties of esential oil.
Author Response
Comments and Suggestions for Authors
Use DES in green extraction process is a novelty and interesting option, in addition, S. chinensis is a promising source of bioactive substances. However, the article requires a wide revision before to be considered for publication. Below more details:
Introduction
- Lines 36-37: Cite more specific reports about uses of DESs for recovery of bioactive compounds from natural sources, particularly antioxidant substances. ¿There is reports on S. chinensis or similar species (e.g. same genere)?
Response: There were few reports on S. chinensis or similar species. Our group previously studied the extraction of six phenolic compounds from rattan (Calamoideae faberii) with deep eutectic solvent. This reference was added as reference [2].
[2] Cao, Q.; Li, J.H.; Xia, Y.; Li, W.; Luo, S.; Ma, C.; Liu, S.X. Green Extraction of Six Phenolic Compounds from Rattan (Calamoideae faberii) with Deep Eutectic Solvent by Homogenate-Assisted Vacuum-Cavitation Method. Molecules 2019, 24, 113. https://doi.org/10.3390/molecules24010113.
- Lines 58-60: The authors cited several properties of S. chinensis, however, only present one reference. In addition, the article cited by the authors (ref number [16]) is a review about antibacterial properties of essential oils from medicinal plants. The authors should present more references relative to the biological properties of S. chinensis.
Response: The reference [16,17] were added to illustrate the biological properties of S. chinensis, such as liver protecting, kidney reinforcing, anti-fatigue, anti-oxidative, immunity enhancing, antiaging, tumor inhibition, and anti-diabetic effects.
[16] Yue, C.J.; Chen, J.; Hou, R.R.; Tian, W.J.; Liu, K.H.; Wang, D.Y.; Lu, Y.; Liu, J.G.; Wu, Y.; Hu, Y.L. The antioxidant action and mechanism of selenizing Schisandra chinensis polysaccharide in chicken embryo hepatocyte. Int. J. Biol. Macromol. 2017, 98, 506-514. https://doi.org/10.1016/j.ijbiomac.2017.02.015.
[17] Cheng, Z.Y.; Yang, Y.J.; Liu, Y.; Liu, Z.G.; Zhou, H.L.; Hu, H.B. Two-steps extraction of essential oil, polysaccharides and biphenyl cyclooctene lignans from Schisandra chinensis Baill fruits. J. Pharmaceut. Biomed. 2014, 96, 162-169. https://doi.org/10.1016/j.jpba.2014.03.036.
- Lines 70-72: Present the suitable references
Response: The references [15] was added in which it studied the ultrasonic-microwave synergistic extraction of polysaccharides from Fortunella margarita (Lour.) Swingle.
[15] Zeng, H.L.; Zhang, Y.; Lin, S.; Jian, Y.Y.; Miao, S.; Zheng, B.D. Ultrasonic–microwave synergistic extraction (UMSE) and molecular weight distribution of polysaccharides from Fortunella margarita (Lour.) Swingle. Sep. Purif. Technol. 2015, 144, 97-106. https://doi.org/10.1016/j.seppur.2015.02.015.
- Lines 61-65: ¿What is the relationship between these sentences and the bioactive properties of S. chinensis?, I do not see it, please improve this paragraph.
Response: The sentence was revised to “and it can prevent cough, promote DNA synthesis and be an inhibitor of plasminogen activation [17]”.
[17] Cheng, Z.Y.; Yang, Y.J.; Liu, Y.; Liu, Z.G.; Zhou, H.L.; Hu, H.B. Two-steps extraction of essential oil, polysaccharides and biphenyl cyclooctene lignans from Schisandra chinensis Baill fruits. J. Pharmaceut. Biomed. 2014, 96, 162-169. https://doi.org/10.1016/j.jpba.2014.03.036.
- Lines 78-81: Move to results and discussion section
Response: Thanks for reviewer’s constructive comment, and the sentence was revised to “The traditional hydro-distillation method was used to extract essential oil, the essential oil was spill over with the boiling water at 100ºC, and The polysaccharide was extracted by aqueous extraction-alcohol precipitation method. Thus, the DES hydro-distillation method was suitable for simultaneously extracting crude polysaccharides and essential oils, and the crude polysaccharide was precipitated by subsequently increasing the alcohol content of the DES solution, and then filtered, which completely separated polysaccharide from DES solution.” in Section 2.7.
Materials and Methods
- Provide more details about sampling process; e.g. maturity of S. chinensis fruits, ¿how much sample was sampling? ¿the sampling was aleatory? Etc
Response: More details about sampling process were added as “Mature fruit picked in September 2017 in the Greater-Lesser Khingan Mountains region of Northeast China” in Section 3.1.
- line 325: ¿how much ChCl and EG were used?
Response: The composition of DES was mixed uniform according to the molar ratio, and the specific molar ratios (1:1, 1:2, 1:3, 1:4, 1:5) were operated and added to the Section 2.1, wherein the optimum molar ratios was 1:3.
- Line 337: This information is not necessary
Response: Thanks for reviewer’s constructive comment, and this information was deleted.
- Line 330: I suggest change the section title, e.g. "Polysaccharide extraction/recovery and analysis". Additionally, present first the extraction procedure (sections 3.3.1.3 and 3.3.1.4), and after the analysis procedure (sections 3.3.1.1 and 3.3.1.2).
Response: The title of Sections 3.3.1.1, 3.3.1.2 and 3.3.1.3 were changed to “Determination method of polysaccharide”, “Determination method of protein”, “Compared extraction method of polysaccharides”, respectively.
- Sections 3.3.1.1 and 3.3.1.2: More specific details about extracts analysis should be cited: e.g. ¿Did you use any preparation procedure for the extracts before analysis them? ¿what amount or concentration of extracts was used? ¿what units used to present the results?
Response: The content “Accurately weighed standard glucose (10.0 mg) into a 100 mL volumetric flask and add water to the scale of 1.0, 2.0, 4.0, 8.0 and 10.0 mL, respectively, to make the volume of the solution 10.0 mL. Take a glucose solution concentration gradient 2.0 mL, and distilled water was mixed to 10.0 mL, with a glucose solution concentration of 2.0 mL gradient, and then add the existing 6% phenol solution 1.0 mL, concentrated sulfuric acid 5.0 mL, shaking and cooling.” was added in Section 3.3.1.1. In addition, the content “Accurately weighed Coomassie brilliant blue G-250 (100.0 mg) was dissolved in 95% ethanol (50.0 mL), 85% phosphoric acid (100 mL) was added, and the mixture was diluted to 1,000 mL with distilled water. Accurately weighed crystal bovine serum protein standard (10.0 mg) was diluted to 100 mL in a volumetric flask with NaCl solution (0.15 M), giving a 0.1 mg/mL albumin standard solution. Add 2.0, 4.0, 6.0, 8.0 and 10.0 mL of NaCl solution (0.15 M) to make the volume of the solution 10.0 mL. Take 0.1 mL of the above albumin concentration gradient solution and add 5.0 mL Coomassie brilliant blue G-250 solution, shaking well.” was added in Section 3.3.1.2.
- Lines 346-347; 355-356; 360-361: ¿temperature?
Response: It is well known that the refluxed temperature of DES was about 100ºC, because in DES, there is large partly water.
- The section 3.3.1.6 is not clear: ¿How the alternating treatments were applied? ¿How the extraction process was optimized? ¿experimental design?
Response: The detailed experimental design process was written in Section 2.3. According to Table 1, the experiments A-G used total polysaccharide extraction time of 60 min. The extraction process was optimized by changing the extraction time and extraction sequence of the ultrasound/microwave under the same conditions.
- Section 3.3.3 ¿what units used to present the results?
Response: The unit of polysaccharide concentration was mg/mL, the unit of cell proliferation rate was %.
- Section 3.3.4 is not clear
Response: The text “The mouse splenic lymphocytes were cultured in DMEM with 10% FBS. Firstly, cell suspensions were seeded in 96-well plates (1 × 105/well), and incubated at 37 °C for 12 h, then the polysaccharides were added. After 6 h, the certain amount of MTT were added into each well and the plate was further incubated for 4 h. Finally, the medium was removed and 200 μL DMSO was added to dissolve the formazan. After 10 min, the absorbance at 490 nm was measured [29].” was added in Section 3.3.4.
[29] Zhang, W.X.; Chen, L.; Li, P.; Zhao, J.Z.; Duan, J.Y. Antidepressant and immunosuppressive activities of two polysaccharides from Poria cocos (Schw.) Wolf. Int. J. Biol. Macromol. 2018, 120, 1696-1704. https://doi.org/10.1016/j.ijbiomac.2018.09.171.
- Section 3.3.5: ¿temperature?
Response: It is well known that the refluxed temperature of DES was about 100ºC, because in DES, there is large partly water.
Results and discussion
- The molar ratios explored were not cite in methods
Response: I’m sorry for my mistake, and the molar ratios explored were added in Section 3.2.
- Line 95: ¿How this optimal condition was established? ¿Experimental design?
Response: The molar ratio of DES was chosen to be ChCl-EG 1:3, and the solution ratios (5:5, 6:4, 7:3, 8:2, 9:1) were changed to obtain an optimal condition, which were added to the Section 2.1. The detailed experimental design process was written in Section 2.3. According to Table 1, the experiments A-G used total polysaccharide extraction time of 60 min. The extraction process was optimized by changing the extraction time and extraction sequence of the ultrasound/microwave under the same conditions.
- Lines 95-98: ¿These results are important? ¿why?
Response: For polyols, there were more than two hydroxyl groups in the molecule that can form large molecules by forming intramolecular hydrogen bonds, and can also form intermolecular hydrogen bonds with ChCl. The greater number of OH was provide by GI, when the same molar ratios of polyol, but the better water absorbent performance of GI may form the intermolecular hydrogen bonds, resulting in a decrease in the number of hydrogen bonds with ChCl. Compared with BDO, the steric effected to the formation of intermolecular hydrogen bonds with was ChCl smaller. Therefore, the screening optimization of DES HBA and HBD was of great significance. It was added in section 2.1.
- After extraction process, ¿how the DESs were recovered?
Response: The polysaccharide was extracted by aqueous extraction-alcohol precipitation method, and DES was dissolved in alcohol solution and removed with liquid phase after 6000 r/min centrifugation. Recovery of partly DES through decompression concentration.
- Section 2.2.1: Kinetic study was not described in the methods section
Response:” Ultrasound-assisted extraction was conducted for 60 min at 550 W, and took a sample every 10 min.” was added in Section 3.3.1.4 and “microwave irradiation was conducted for 30 min at 250 W, and took a sample every 5.0 min.” was added in Section 3.3.1.5.
- Line 111: ¿optimal extraction times? I suggest not use the word "optimal". In this work the authors did not optimize the extraction process.
Response: The highlight of this study was the optimization of the synergistic/alternative process of ultrasonic and microwave. Therefore, the single factor analysis of the extraction process (Table 1) in the early stage provided relatively optimized parameters for the subsequent experiments. So, the word "optimal" was used for alternative process.
Table 1. Experimental program of alternating ultrasound/microwave extractions of polysaccharides.
Item | Experimental program | Essential oil yield (mL/kg) | Polysaccharide yield (g/100 g) | Environmental impact (g CO2 rejected) |
A | UAE 60 min | 10.6 | 7.02 ±0.11 | 440 |
B | MAE 60 min | 10.9 | 7.83 ±0.12 | 200 |
C | UAE 40 min + MAE 20 min | 11.2 | 8.19 ±0.20 | 360 |
D | MAE 20 min + UAE 40 min | 11.5 | 8.48 ±0.33 | 360 |
E | UAE 20 min + MAE 10 min + UAE 20 min + MAE 10 min | 11.6 | 8.50 ±0.31 | 360 |
F | MAE 10 min + UAE 20 min + MAE 10 min + UAE 20 min | 12.2 | 8.87 ±0.29 | 360 |
G | UAE & MAE 60 min | 11.9 | 8.79 ±0.26 | 640 |
H | Reflux extraction 4 h | 11.7 | 8.56 ± 0.30 | 3200 |
- Section 2.2.1 and 2.2.2: The experimental design to optimize the extraction process is not clear. For RE the authors “optimized” each variable independently, however they not consider the interactions between variables (e.g. time interaction with solid-liquid ratio)
Response: The highlight of this study was the optimization of the synergistic/alternative process of ultrasonic and microwave. The single factor analysis of the extraction process (Figure 2, 3) in the early stage provided relatively optimized parameters for the subsequent experiments. Therefore, the interactions between variables of this study was not in a single process.
- Section 2.2.3: ¿What is the interaction between the extraction time and the power?
Response: The highlight of this study was the optimization of the synergistic/alternative process of ultrasonic and microwave. The single factor analysis of the extraction process (Figure 2, 3) in the early stage provided relatively optimized parameters for the subsequent experiments. The extraction time of ultrasonic or microwave was optimum in alternative test design again. Therefore, the interaction between the extraction time and the power was not considered in single factor test of single process.
- Sections 2.2: Is necessary discuss results
Response: The optimum of ultrasonic and microwave power and extraction time was used for alternative experimental design. Therefore, it was the necessary discuss results.
- Section 2.10 should be move at section in order to explain the alternate UA/M extraction process
Response: Thanks for reviewer’s constructive comment, and Section 2.10 was moved to Section 2.3.
- Figure 6: ¿What polysaccharides extract was used?
Response: “The different concentration of polysaccharide extract was 0.5, 0.4, 0.3, 0.2, 0.1 mg/mL, respectively”, and it was added in Section 2.5.
- Explain how the polysaccharides scavenging the OH* radical
Response: “The hydroxyl radicals were produced by Fenton reaction. Hydrogen peroxide/Fe2+ system can produce free radicals as follows: H2O2 + Fe2+ → •OH + OH– + Fe3+. It can be seen that the generation of a hydroxyl radical depends on the content of Fe2+ and H2O2. The polysaccharide should have scavenging effect on reactive oxygen free radicals, which could scavenge the excessive oxygen free radicals produced in vivo, block the free radical reaction chain in vivo;” “Moreover, there were several nature metal ions (such as Fe, Zn, Cu) in polysaccharide, which can also promote free radical scavenging.” This explanation was added in Section 2.5.
- Section 2.7: similar to the comments of section 2.2.1
Response: The traditional hydro-distillation method was used to extract essential oil, the essential oil was spill over with the boiling water at 100ºC, and The polysaccharide was extracted by aqueous extraction-alcohol precipitation method. Thus, the DES hydro-distillation method was suitable for simultaneously extracting crude polysaccharides and essential oils, and the crude polysaccharide was precipitated by subsequently increasing the alcohol content of the DES solution, and then filtered, which completely separated polysaccharide from DES solution. This process was indeed the same, the crude polysaccharides and essential oil were extracted in one step.
- Section 2.8: results requires discussion
Response: Thanks for reviewer’s constructive comment, the results of "the antioxidant activity of essential oils have shown that monoterpenes have a higher antioxidant effect, while the major component of S. chinensis essential oil were sesquiterpenes. Therefore, in my opinion, the lack of monoterpenes in essential oil was one of the possible reasons for its weak antioxidant activity [22]." This explanation were discussed and added in Section 2.8.
[22] Hasheminya, S.M.; Mokarram, R.R.; Ghanbarzadeh, B.; Hamishekar, H.; Kafil, H.S.; Dehghannya, J. Development and characterization of biocomposite films made from kefiran, carboxymethyl cellulose and Satureja Khuzestanica essential oil. Food. Chem. 2019, 18. https://doi.org/10.1016/j.foodchem.2019.03.076.
- Section 2.9: ¿What is the relationship between chemical composition and the observed antioxidant properties? The Chemical composition should be related with the biological properties of esential oil.
Response: “The olefin had high unsaturation and strong reducing ability. As can be seen from Figure 11, the olefin content was relatively high, such as Ylangene (RA=28.63%), (E)-α-bergamotene (RA=10.62%), and β-himachalene (RA=10.60%), thus the essential oil had a certain antioxidant capacity.” This explanation was added in Section 2.9.
Main components of essential oils:
Ylangene
(E)-α-bergamotene
β-himachalene

Round 2
Reviewer 1 Report
Line 103/104 - This sentence makes no sense.
Line 255/256 . If the authors did not determine the concentration of the metal ions, they cannot claim that their conc.was higher in some samples, they can only assume it. Furthermore, can they correlate the metal ions conc.with the extraction parameters? If no, it should be written as the assumption.
the authors claim: he traditional hydro-distillation method was used to extract essential oil, the essential oil was spill over with the boiling water at 100ºC, and The polysaccharide was extracted by aqueous extraction-alcohol precipitation method. Thus, the DES hydro-distillation method was suitable for simultaneously extracting crude polysaccharides and essential oils, and the crude polysaccharide was precipitated by subsequently increasing the alcohol content of the DES solution, and then filtered, which completely separated polysaccharide from DES solution.” in Section 2.7.
It is still not clear how the application of DES influenced the higher yield of the essential oil. Is it because of the hydrogen bonding which causes the higher boiling point or something else? It should be explained in detail. What was the initial assumption for using DES for hydro-distillation?
The English language should be improved.
Author Response
Line 103/104 - This sentence makes no sense.
Response: Thanks for reviewer’s constructive comment, and this information was deleted. Line 255/256. If the authors did not determine the concentration of the metal ions, they cannot claim that their conc. was higher in some samples, they can only assume it. Furthermore, can they correlate the metal ions conc. with the extraction parameters? If no, it should be written as the assumption. Response: The reference was added as reference [22,23].
[22] Zhang, Z.S.; Wang, X.M.; Zhao, M.X.; Qi, H.M. Free-radical degradation by Fe2+/Vc/H2O2 and antioxidant activity of polysaccharide from Tremella fuciformis. Carbohyd. Polym. 2014, 112, 578-582. https://doi.org/10.1016/j.carbpol.2014.06.030.
[23] Xiong, X.; Huang, G.L.; Huang, H.L. The antioxidant activities of phosphorylated polysaccharide from native ginseng. Int. J. Biol. Macromol. 2019, 126, 842-845. https://doi.org/10.1016/j.ijbiomac.2018.12.266.
the authors claim: The traditional hydro-distillation method was used to extract essential oil, the essential oil was spill over with the boiling water at 100ºC, and the polysaccharide was extracted by aqueous extraction-alcohol precipitation method. Thus, the DES hydro-distillation method was suitable for simultaneously extracting crude polysaccharides and essential oils, and the crude polysaccharide was precipitated by subsequently increasing the alcohol content of the DES solution, and then filtered, which completely separated polysaccharide from DES solution.” in Section 2.7.
It is still not clear how the application of DES influenced the higher yield of the essential oil. Is it because of the hydrogen bonding which causes the higher boiling point or something else? It should be explained in detail. What was the initial assumption for using DES for hydro-distillation?
Response: Thanks for reviewer’s constructive comment, and the text “As for ChCl based acidic DESs, studies have inferred that Cl- of ChCl (HBA) with strong H- bonds acceptability could help to disrupt the intermolecular hydrogen bonding network of biomass and facilitate it dissolution, which was benefit for the subsequent HBD access and attack on the acid catalytic sites of biomass, thus better biomass deconstruction and hydro-distillation performance of DESs solution was achieved as compared to the water, thereby increasing the yield of the essential oil [24].” was added in Section 2.7. DES were easy to be prepared not requiring purification steps, were made from low cost compounds, present low or negligible toxicity and were biodegradable and easily recyclable. These features made DES preferable over the conventional solvents used in extraction procedures, so we chose to use DES for hydro-distillation
[24] Li, A.L.; Hou, X.D.; Lin, K.P.; Zhang, X.; Fu, M.H. Rice straw pretreatment using deep eutectic solvents with different constituents molar ratios: Biomass fractionation, polysaccharides enzymatic digestion and solvent reuse. J. Biosci. Bioeng. 2018, 126, 346-354. https://doi.org/10.1016/j.jbiosc.2018.03.011.
The English language should be improved.
Response: Yes,we checked all text carefully.
Reviewer 2 Report
Introduction
It is suitable
Materials and Methods
-About authors reply, I suggest to added the information "The composition of DES was mixed uniform according to the molar ratio, and the specific molar ratios (1:1, 1:2, 1:3, 1:4, 1:5)...... " in section 3.2
-Section 3.3.1.6: To present the experimental details in M&M section
-Section 3.3.3: According to authors reply, please cited the units in M&M section
Results and Discussion
-In the revised version of manuscript, the molar ratios explored were not cite in methods, although the authors replied "molar ratios explored were added in Section 3.2."
- Table 1 and the experimental design should be described in the M&M section
- The response to the question, "how the DES were recovered?" is suitable, please include in M&M section
- Section 2.2.1: Following the authors response, I suggest add in the sections 3.3.1.4 and 3.3.1.5 the sentence "took samples every 10 min in order to obtain the extraction kinetic curve.... "
- Section 2.2.1 to 2.2.3: The authors reply is not satisfactory.
